# An Extended Theory of Planned Behavior to Explain General Contractors' Long-Term Cooperation Intentions in Construction Projects: Empirical Evidence from China

**Xun Liu** [1,*] , **Dexin Liu** [2] **and Mengyu Xu** [2]

1   School of Civil Engineering, Suzhou University of Science and Technology, Suzhou 215000, China
2   School of Business, Suzhou University of Science and Technology, Suzhou 215000, China
*   Correspondence: liuxun8127@usts.edu.cn

**Abstract:** With the continuous development of the construction industry, the current construction production mode is gradually transforming into the contractor, professional subcontractor and labor subcontractor's "main-sub contractor" cooperative production mode in the market segmentation. Long-term cooperation between contractors and subcontractors is beneficial to both parties, yet only limited research has explored the factors that determine contractors' willingness to cooperate with subcontractors on a long-term basis. This paper explores the factors that influence contractors' willingness to cooperate with subcontractors in the long term and the forming mechanisms. To achieve this goal, this study combines the characteristics of the construction industry to increase the variable of past experience, constructing an extended planned behavior theory model. Through questionnaire surveys, contractors with experience in subcontract management were surveyed, and the hypotheses were tested using structural equation modeling techniques. The results showed that contractors' long-term cooperation intention was mainly influenced by attitudes, subjective norms, and perceived behavioral control. While past experience, a new variable, had significant effects on attitudes and perceived behavioral control, influencing contractors' long-term cooperation intention by affecting attitudes and perceived behavioral control, past experience had no direct effect on subjective norms. This study will explain the formation mechanism of the general contractor and subcontractor's long-term cooperative relationship and provide a theoretical basis for the general contractor to select partners and suggestions for subcontractors to improve their work.

**Keywords:** organization; theory of planned behavior; collaboration; construction industry

## 1. Introduction

Construction has always been considered one of the country's most important sectors. Large-scale construction projects are complex and multidisciplinary. Under impacts in the big picture, the traditional management mode is increasingly restricting the development of construction enterprises due to fierce competition and professional differentiation [1]. As traditional labor-intensive enterprises, it is becoming increasingly difficult for any construction company to survive alone. As construction projects are often large and complex, contractors often need to integrate subcontracting of effective resources to complete some parts of the project [2,3]. Therefore, establishing a sustainable working relationship between project managers and subcontractors, identifying the root causes of key obstacles, and addressing and dealing with them through the use of all necessary techniques is key to project success. It is an effective way to create a sustainable cooperation relationship for pooling finances, technology, skills and specializations [4,5].

Due to the requirements of specialization, resource integration and risk dispersion, for a large and complex project the number of contract packages divided by contractors can be as many as dozens or even hundreds [6,7]. With the differentiation of specialties, the management mode between main contracting and subcontracting is also changing

gradually [8,9]. The reasons are as follows: From the perspective of government, the state has vigorously advocated the contracting model in recent decades. Increasingly, Engineering Procurement Construction (EPC) and Build-Operate-Transfer (BOT) projects are being undertaken on the market [1,10]. For the corresponding policy call and market demand, the project delivery model through mutual cooperation between contractors, professional subcontractors and labor subcontractors has gradually formed [11,12]. From the perspective of market demand, with regard to projects that are more and more complex, larger and larger, and have more and more strict construction-period requirements, it is often difficult for contractors to successfully complete the delivery by themselves [13]. For projects with complex construction techniques and high requirements for professional levels, contractors may not have the corresponding construction capabilities [2,14]. Thus, they need to invest a lot of manpower and material resources, and the "input-output" ratio is not high [13,15]. Therefore, the contractors often transfer this part of work through subcontracting and improve their own core competitiveness through subcontracting management and coordination. From the perspective of major project stakeholders, the owners have higher requirements for project quality and degree of specialization [16,17]. Some technical problems in the project must be completed by professional subcontracting. To a greater extent, as for the contractor, transferring risks and project workload through subcontracting can reduce their own burden, streamline the project team and protect their own interests. For professional subcontractors and labor subcontractors, focusing on a specific segment can improve their market position and core competitiveness in this area [3,18]. Through the influence of the above three points, the construction industry has gradually changed from the traditional mode to the contractor-led project delivery management mode marked by cooperation between the general contractor and subcontractors [1,19,20].

The importance of the relationship between general contractor and subcontractor in engineering projects is becoming more and more prominent, and many scholars have studied it. However, while most studies focus on the cooperative behavior of subcontractors [21,22], it is important to note that the cooperative behavior is largely determined by the cooperative intention. Therefore, it is necessary to pay attention to the long-term cooperation intention of general contractors and subcontractors. Since the general contractor plays a dominant role in the sustainable cooperative relationship between the general contractor and subcontractor, this study intends to be grounded on the general contractors' position to develop such a relationship. As for the initial cooperation of general contractor and subcontractor, past experiences involving reputation and workability will provide more information and a basis for the general contractor to select subcontractors. [23,24]. Using the extended theory of planned behavior (TPB) as a theoretical framework, this study explores what factors affect the contractor's choice of long-term cooperation and how to achieve the cooperation intention with subcontractors [25,26].

From the perspective of theoretical significance, this study first improves the existing literature by comprehensively identifying the factors that affect the long-term cooperation intention of contractors and subcontractors. Second, based on the TPB, an influence model is constructed to enrich the theoretical system of subcontractors and contractors. Practically, this study was useful in helping the general contractor select a partner, while also providing suggestions for subcontractors to improve their work.

The remaining sections of this paper are organized as follows. Section 2 offers a review of the literature on relationships between general contractors and subcontractors and the TPB. Section 3 describes the theoretical framework and research hypotheses. Section 4 describes the design of the survey questionnaire and data collection. The results of the data analysis are presented in Section 5. Section 6 addresses the key findings from this analysis and research deficiencies. In the end, conclusions are presented along with future research directions.

## 2. Literature Review

### 2.1. Relationships between General Contractors and Subcontractors

With the importance of the relationship between general contractors and subcontractors in project collaboration being highlighted, business and academia gradually realized that a hostile relationship between construction participants will affect project cost, schedule, quality, etc., and potentially lead to disputes, claims and litigation, etc. Hence, construction participants converted into an interest community can achieve greater project success [3]. Therefore, more and more experts and scholars have begun to study main contract and subcontract cooperation [27,28]. Numerous interface factors may influence the contractor–subcontractor relationship throughout the entire construction process. There are many factors that contribute to this result, including low levels of conflict, mutual trust, effective coordination, and open communication [29–31].

In terms of the contractual relationship that exists between the general contractor and the subcontractor, the subcontract defines the rights, obligations and responsibilities of both parties, as well as the risk-sharing mechanism [2,32,33]. Additionally, in the case of subcontracts, the subcontractor completes a specialized portion of the work for the general contractor. The latter is responsible for (1) managing subcontractors and ensuring that their construction operations satisfy the specifications of the client, as well as (2) coordinating with a group of subcontractors. In some way, the general contractor and several subcontractors are interdependent [34,35]. Most general contractors adopt three types of relationships between them and their subcontractors, namely, long-term business relationships, short-term business relationships, and hostile-dependent relationships.

Long-term partnerships can improve performance and significantly improve construction project delivery [34]. To begin with, in order to finish each major trade of construction work, such as excavating, drainage laying, formwork, concreting, metalwork, etc., many general contractors work with a cluster of subcontractors in a long-term cooperative relationship, which may last for more than 10 years in a conservative estimate [9]. Cooperation between general contractor and subcontractor results in positive performance records that benefit both parties. In other words, if they work together as long-term partners or teammates, both parties stand to achieve [36]. In addition, many general contractors maintain one-time or short-term business relationships with subcontractors who have the specific capabilities or qualifications to perform specialized construction work to a high standard [37]. Nevertheless, if the general contractor is satisfied with the construction performance and the level of cooperation, there will be a long-term intention to cooperate [38]. Long-term cooperation intention refers to the behavioral orientation or ideological tendency of both parties to hold a long-term joint undertaking. In the area of engineering construction, many scholars have investigated the cooperation intention of main contractors and subcontractors [39,40]. Moreover, in a hostile-dependent relationship that can develop between general contractors and subcontractors, both are mainly concerned with their self-interest [41]. Considering the complexity of construction projects, the entry of new competitors, and the involvement of new stakeholders in the construction process, such hostile dependencies are often adversarial and urgent [42–44]. This study focuses on the long-term cooperation intention and defines it as the tendency of the contractor to cooperate with the subcontractor for a long time.

### 2.2. Theory of Planned Behavior

This study uses the Theory of Planned Behavior (TPB) to investigate the factors influencing behavioral intentions and the relationship between behavior and intentions. The TPB argues that an individual's intention to perform a behavior is determined by a number of factors, including positive evaluations of the behavior (attitudes), social pressure to encourage the behavior (subjective norms), and perceived ease of the behavior (perceived behavioral control) [45]. In accordance with TPB's recommendations, it is widely used to explain and anticipate behavior. It is also commonly used in management and social sciences to study the effects of subjective psychological factors on individual behavior.

Behavior intention (BI) is a psychological factor that influences behavior directly and reflects motivation for performing a particular action. It is more likely that a behavior will be implemented if it has a strong intention. BI is ruled by three predetermined variables: subjective norms (SN), attitudes towards the behavior (AB), and perceived behavioral control (PBC) [46]. SN is an individual perception of external social pressure to perform the behavior; AB is a positive or negative evaluation of the performance of a behavior based on individual strengths; PBC is the assessment by an individual of the external resources and conditions under which the behavior is to be implemented [47].

Rational behavior theory and planned behavior theory all pay close attention to analyzing the measurement of subjective feelings of interviewees, rather than the measurement of objective data or observation [48]. Many methods have been used to quote the TPB, which was proposed in the field of social behavior many years ago. Some scholars have conducted empirical validation, such as green behavior, online shopping behavior, product choice, and eating behavior, which have been well verified [49–51]. Such validation shows that empirical studies have good predictive power for intention and behavior.

However, the continuous application and promotion of planned behavior theory has caused some scholars to question it. They argue that attitudes, perceived behavioral control, and subjective norms cannot fully cover and explain the intentions and behaviors of people in all domains. Apart from these three variables, many behaviors are also affected by other factors [47,52,53]. For instance, in the field involving past experience, the past behavior should be added as an important variable, while in the field related to integrity, moral hazard or moral concept is an important factor. The introduction of two variables, emotion and moral perception, gives the extended TPB greater predictive power [54]. A growing number of scholars have used the planned behavior theory model according to the characteristics of the object of study. The theoretical model is extended by adding different measured variables that fit the model context. In this way, the predictive power of the model is improved, as is the degree of explanation for intentions and behaviors [45,55]. Using the TPB, this study constructs a model of contractors' long-term cooperative intention based on the process of establishing the model.

## 3. Hypotheses and Theoretical Framework

This study used TPB as a theoretical framework to characterize the long-term cooperation intention of general contractors and subcontractors. Based on the original attitude (AB), perceived behavior control (PBC) and subjective norms (SN), this study increases the variable of past experience (PE), so as to propose the expanded planned behavior theory model and to explore the relationship between these four variables and long-term cooperation intention between contractor and subcontractor.

### 3.1. Long-Term Cooperation (LTC)

Intention represents the motivation to implement a certain behavior. In this study, long-term cooperation intention refers to the degree to which the relationship with subcontractors is to be established and pursued. Therefore, a long-term cooperation intention is different from long-term cooperative behavior. Intention can largely determine behavior, while long-term cooperative behavior is determined by many uncontrollable factors owing to the generation of subcontractors through the bidding process. The intention to pursue a long-term sustainable cooperation relationship is considered from the following three aspects: the hope of maintaining long-term cooperation, preferential terms, and investment for more resources in future [5,35,56].

### 3.2. Planned Behavior Theory Original Variables
#### 3.2.1. Attitude towards Behavior (AB)

Attitude towards behavior involves the positive or negative evaluation of an individual for performing a particular behavior [57,58]. Evaluation of things alone would not affect their intention to carry out the behavior, but a positive or negative evaluation would

affect their intention to do so. When the evaluation is positive, the individual is more inclined to put the behavior into effect; otherwise, when the evaluation is negative, the individual intention to carry out the behavior is weakened [59]. In the engineering industry, the intention of a contractor to cooperate with a subcontractor on a long-term basis is also influenced by contractor's attitude towards the execution of the contract. The attitude of general contractors towards subcontractors refers to the attitude of general contractors regarding the long-term cooperation with subcontractors, that is, the attitude of general contractors regarding the implementation of such behavior. The long-term cooperation between general contractors and subcontractors is a subjective attitude of general contractors on whether to maintain long-term cooperation with subcontractors after general contractors have integrated relevant information about subcontractors. In this study, the attitude of general contractors towards subcontractors is measured by five types of items: progress requirements; quality requirements; good brand image; consolidation of local market and increase of share; confidence and commitment [3,8]. Thus, on the basis of the above analysis, Hypothesis 1 was proposed:

**Hypothesis 1.** *AB has a positive influence on the intention for a sustainable cooperation relationship.*

### 3.2.2. Subjective Norms (SNs)

Subjective norms are determinants of personal norms. In addition to confirming the social correctness of specific behaviors, SNs can assist individuals in determining if their beliefs and norms align with those of others. Several studies have demonstrated that SNs play a substantial role in affecting behavioral attitudes [46,47,60]. In terms of the construction engineering, the social correctness of the intention to create a sustainable cooperation relationship is considered based on five aspects: contract requirements; local laws and regulations; general contractors' specific requirements; clients' specific requirements; other similar projects [3,61]. In light of the previous analysis, Hypothesis 2 was proposed:

**Hypothesis 2.** *SNs have a positive influence on the intention for sustainable cooperation relationship.*

### 3.2.3. Perceived Behavioral Control (PBC)

PBC is an unconscious factor reflecting perceptions of how comfortable or difficult it is to put a particular behavior into practice. In general, the simpler the behavior appears to be compared to individual ability, the more likely they are to engage in it. In contrast, the more difficult the comparison, the less likely they are to engage in such a behavior. Therefore, PBC has a positive relationship with intention [47,51,62,63]. For general contractors, PBC refers to the cognition of self-ability in a long-term cooperation with subcontractors and the ability to build a good long-term cooperative relationship with subcontractor. Meanwhile, it is usually considered from the perspectives of interface division, power, responsibility and benefits division, communication ability, subcontract management, etc. [3,61]. Based on the above discussion, the following Hypothesis 3 was proposed:

**Hypothesis 3.** *PBC has a positive influence on the intention for sustainable cooperation relationship.*

### 3.3. Extended Planned Behavior Theory Variable—Past Experience

Throughout the last few decades, researchers have sought to improve the application degree and explanatory abilities of the TPB model through changing approaches and/or adding new variables [51,53,64]. Ajzen [26] showed that two points should be noted in the extension of the theory of planned behavior: (1) the newly added variables should be clearly defined and distinguished from the original variables, and (2) the newly added variables should be matched with the original variables. Though TPB has been put into practice in various backgrounds successfully, e.g., shared bicycles [51], e-learning [65], autonomous vehicle [50], online shopping [66], waste management [67], etc., it seems that hardly any research has been implemented to apply the theory to sustainable cooperative

intention relationship among contractors. This study tried to extend the TPB model by adding new variables (past experience) and modifying the approach in the model to predict a formation mechanism of long-term cooperative intention for general contractors.

As a significant factor in decision-making processes, past experience is considered an influential determinant of attitudes and perceived behavioral control [68,69]. In this study, the process of choosing long-term cooperative subcontractors is a process of continuous adjustment based on past experience, which includes analyzing internal and external needs, establishing goals, optimizing relations constantly, and conducting pre- and post-event evaluation [2,70,71]. Considering the above literature, there is a basis to be able to integrate past experience into a model which aims to explain long-term cooperative intention formation. Thus, the Hypotheses 4 to 6 were proposed:

**Hypothesis 4.** *PE have a positive influence on the intention for sustainable cooperation relationship.*

**Hypothesis 5.** *PE have a positive influence on SNs for sustainable cooperation relationship.*

**Hypothesis 6.** *PE have a positive influence on PBC.*

To sum up, Figure 1 depicts the extended TPB model. It includes the original variables (attitude, subjective norm, and perceived behavioral control) as well as novel constructs (past experience). The thick lines represent the new paths added to the original TPB model.

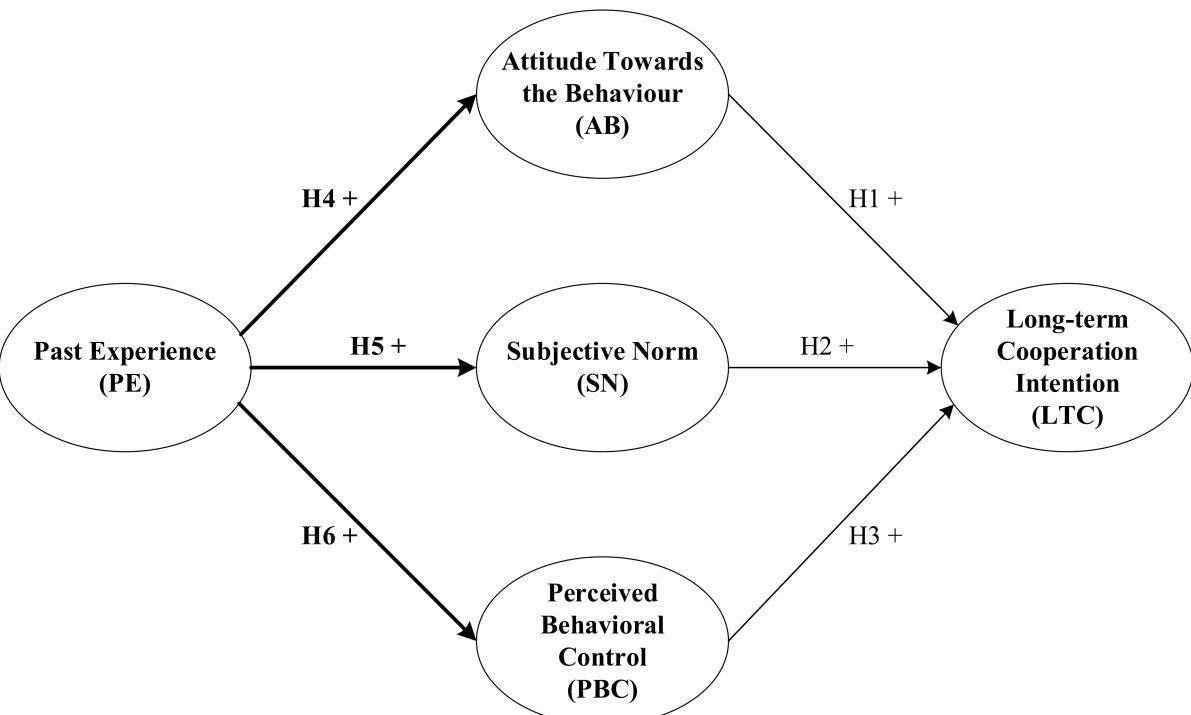

**Figure 1.** Extended planned behavior theory of long-term cooperation intention.

## 4. Methodology

Structural equation modeling (SEM) is used to test the proposed hypotheses. SEM analysis is a measurement technique that enables complex causal links between sample data to be expressed in terms of the corresponding model equations and to be measured and analyzed. SEM is therefore particularly suitable for testing predictive and extended theoretical models. In this study, a two-step testing approach was used to first assess the reliability and validity of the measurements included in the current study and then test the hypothesized relationships.

### 4.1. Questionnaire Design

This study intended to collect data by combining online and offline questionnaires. In order to ensure the questionnaires' quality, a pilot study was conducted by inviting ten experts working as general contractors before beginning a large-scale distributing of questionnaires. The preliminary questionnaire can determine whether the indicators are in line with an actual situation and miss or repeat questions. The formal questionnaire includes three parts: (1) background information of respondents, including their personal project experience; (2) background information of cooperation experience, which mainly includes length and number of cooperation, etc.; respondents were asked to answer the questionnaire considering the background of the subcontractor in the most recent project they were involved in; (3) the measurement of the general contractors' long-term cooperation intention using each variable in the expanded planned behavior theory.

Based on the context of the general contractors' intention to foster a sustainable cooperative relationship with subcontractors, Table 1 displays the measurement methods for each variable. The questionnaire consists of five constructs and 22 scale items, including five measures of attitude, five measures of subjective norms, five measures of perceived behavioral control, four measures of past experience, and three measures of long-term cooperative intention. A five-point Likert scale was used to assess the items, and respondents were asked to give a rating of 1 to 5 on each item (1 = strongly disagree, 3 = agree, and 5 = strongly agree).

**Table 1.** Sources of constructs and items used in the study.

| Constructs | ID | Measuring Item | Source |
|---|---|---|---|
| **Attitude towards the behavior (AB)** | AB1 | Subcontractor can meet progress requirements of general contractor | [3,8,61,72] |
| | AB2 | Subcontractor can meet quality requirements of general contractor | |
| | AB3 | Subcontractor can help general contractor establish favorable brand image | |
| | AB4 | Cooperating with subcontractors can help general contractor consolidate local market, increasing market share | |
| | AB5 | Subcontractor can achieve trust and commitment with general contractor | |
| **Perceived behavioral control (PBC)** | PBC1 | General contractor has sufficient ability to clear role positioning and interface division by cooperating with subcontractors. | [3,8,61,72] |
| | PBC2 | General contractors have sufficient ability to share risks and benefits with subcontractors. | |
| | PBC3 | General contractors have sufficient ability to communicate and cooperate with subcontractors. | |
| | PBC4 | General contractors have sufficient capacity to manage subcontractors. | |
| | PBC5 | General contractors have sufficient capacity to ensure project successful by cooperating. | |
| **Subjective norm (SN)** | SN1 | Cooperation with subcontractors is conforming to the law. | [3,8,61,72] |
| | SN2 | Subcontractors have qualified engineering projects qualifications. | |
| | SN3 | Other similar projects establish a good relationship with subcontractors. | |
| | SN4 | General contractor company puts forward requirements for selecting subcontractors. | |
| | SN5 | The proprietor puts forward requirements for the selection of subcontractors. | |

**Table 1.** *Cont.*

| Constructs | ID | Measuring Item | Source |
|---|---|---|---|
| **Past experience (PE)** | PE1 | Subcontractor has a good reputation and performance in the industry. | [2,69,70,73,74] |
| | PE2 | Subcontractor has completed work well in the past experience. | |
| | PE3 | Subcontractor has always been able to give good suggestions that brought additional benefits to general contractors' projects. | |
| | PE4 | Subcontractor is similar to general contractors' corporate organizational culture. | |
| **Long-term cooperation intention (LTC)** | LTC1 | General contractors hope to maintain long-term cooperation with subcontractor | [5,8,9,35,75] |
| | LTC2 | General contractors are willing to give preferential conditions | |
| | LTC3 | General contractors are willing to invest more resources to maintain long-term cooperative relationship in future cooperation | |

*4.2. Sampling and Data Collection*

This study intends to explore the long-term cooperative intention between contractor and subcontractor. Directional screening of respondents can ensure the availability of questionnaire data. Therefore, this study issued questionnaires to general construction contractors with contracting experience. Between June 2021 and December 2021, 215 questionnaires were collected, of which 165 were considered valid. In this instance, the logical errors and questionnaires with invalid results have been eliminated and cleaned. The questionnaire validity rate was 76.74%. The participant's IP address is limited to a single submission in order to prevent the same person from submitting the same questionnaire more than once.

**5. Data Analysis**

*5.1. Demographics and Descriptive Findings*

The sociodemographic characteristics of this survey are shown in Table 2. It can be observed that the education level of the respondents is relatively high: More than three quarters had a bachelor's degree or higher. Meanwhile, more than 75 percent of the respondents had worked in the construction industry for more than five years. The majority of respondents were in the position of general manager or above. In addition, more than half of the respondents participated in three or more construction projects as main contractors.

**Table 2.** Statistical analysis of questionnaire sources.

| Feature | Classification | Quantity | Proportion |
|---|---|---|---|
| Education | Junior college and below | 38 | 23.0% |
| | Bachelor | 69 | 41.8% |
| | Master and above | 58 | 35.2% |
| Work experience | 1–5 years | 43 | 26.1% |
| | 6–10 years | 42 | 25.5% |
| | 10–15 years | 45 | 27.3% |
| | Over 16 years | 35 | 21.2% |
| Management position | Senior manager | 39 | 14.7% |
| | General manager | 68 | 41.2% |
| | Technical personnel | 58 | 35.2% |
| Number of projects participating as main contractors | 1–3 | 72 | 43.6% |
| | 4–6 | 58 | 35.2% |
| | 7–9 | 25 | 9.4% |
| | Over 10 | 10 | 6.1% |

### 5.2. Reliability and Validity Tests

The models and hypotheses proposed in this study were tested using SPSS and AMOS. Before doing the hypothesis test, it is inevitable to assess the fitness of the model. The measurement model's reliability and validity should be examined by applying Confirmatory factor analysis (CFA). A check of the internal consistency of the items in each variable should be made by using Cronbach's alpha values and composite reliability (CR) [76]. As Table 3 shows, Cronbach coefficients of the five groups of variables are all 0.7 above the consistency test threshold, indicating that the measurement model has been proved to meet the reliability requirements through reliability [77].

**Table 3.** Reliability test.

| Group | Construct | Cronbach'$\alpha$ |
|---|---|---|
| Attitude towards the behavior | AB | 0.779 |
| Perceived behavioral control | PBC | 0.734 |
| Subjective norms | SN | 0.768 |
| Past experience | PE | 0.801 |
| Long-term cooperation intention | LTC | 0.809 |

Convergent validity is an order to test the significance of different observed variables of the same latent variable to determine the closeness of each observed variable, so as to judge whether each observed variable can measure a latent variable accurately. The criteria for passing the convergent validity are factor loading $\geq$ 0.45; combined reliability $\geq$ 0.7; average variance extraction $\geq$ 0.5. As shown in Table 4, factor loading, average variance extraction, and combined reliability all meet the threshold requirements; it therefore follows that this model passes the convergent validity test.

**Table 4.** Convergent validity test.

| Group | No. | Factor Load | AVE | CR |
|---|---|---|---|---|
| | AB1 | 0.525 | 0.597 | 0.833 |
| | AB2 | 0.832 | | |
| AB | AB3 | 0.856 | | |
| | AB4 | 0.823 | | |
| | AB5 | 0.811 | | |
| | PBC1 | 0.678 | 0.765 | 0.845 |
| | PBC2 | 0.751 | | |
| PBC | PBC3 | 0.863 | | |
| | PBC4 | 0.589 | | |
| | PBC5 | 0.605 | | |
| | SN1 | 0.512 | 0.614 | 0.819 |
| | SN2 | 0.745 | | |
| SN | SN3 | 0.781 | | |
| | SN4 | 0.746 | | |
| | SN5 | 0.691 | | |
| | PE1 | 0.611 | 0.589 | 0.766 |
| PE | PE2 | 0.592 | | |
| | PE3 | 0.746 | | |
| | PE4 | 0.721 | | |
| | LTC1 | 0.811 | 0.834 | 0.841 |
| LTC | LTC2 | 0.912 | | |
| | LTC3 | 0.872 | | |

In contrast to the convergent validity test, the discriminate validity test is aimed at demonstrating the degree of discrimination between different latent variables. By measuring the correlation significance between the observed variables of different latent variables, we can judge whether there is a distinction difference between the various latent variables. The square root of each latent variable's AVE value must be better than or equal to the coefficient between each latent variable in order to meet the test criterion for discriminate validity. If it meets that criterion, it means that the latent variable meets the test of discriminate validity.

According to Table 4, each value under the diagonal line corresponds to the correlation coefficient between the corresponding two latent variables, and the value above the diagonal line is the square root of the AVE of each latent variable. The data in Table 5 demonstrate that the square root of AVE is higher than the correlation coefficient of each latent variable, demonstrating the high discriminative validity of the five latent variables. The reliability and validity tests show that the model has ideal reliability and validity to measure the path of the structural model.

**Table 5.** Differential validity.

| Group | AT | PBC | SN | PE | IN |
|:-----:|:----:|:----:|:----:|:----:|:----:|
| AT | 0.779 | | | | |
| PBC | 0.453 | 0.878 | | | |
| SN | 0.464 | 0.345 | 0.723 | | |
| PE | 0.272 | 0.224 | 0.235 | 0.871 | |
| LTC | 0.592 | 0.545 | 0.425 | 0.288 | 0.751 |

*5.3. Structural Model and Hypothesis Tests*

A bootstrapping algorithm is suitable for a small sample size and can improve the accuracy and reliability of results of structural equation model by repeated sampling. This paper adopted a bootstrapping algorithm and set 5000 as the new sample size for repeated sampling. When the significance level was 0.05, the *t*-value of a two-sided test should be greater than 1.69, indicating that the data support this hypothesis. The calculation results of the bootstrapping algorithm are shown in Table 6, where the *t*-value of the five paths H1, H2, H3, H4 and H6 are all above 1.69, demonstrating that the path coefficients of the five groups are positive and important. It also indicates that all five hypotheses are valid at the 0.05 level of significance. The *t*-value of H5 is 1.235, less than 1.69, indicating that the H5 hypothesis is not valid. In general, five out of the six hypotheses are valid and the results of the inner model are good. Therefore, the inner model is valid.

**Table 6.** Path coefficient and significance level.

| Hypothesis of Path | Path Coefficient | *t*-Value | Explanation |
|:------------------:|:----------------:|:---------:|:-----------:|
| H1: AB→LTC | 0.447 | 5.727 | Supported |
| H2: SN→LTC | 0.328 | 4.331 | Supported |
| H3: PBC→LTC | 0.542 | 6.972 | Supported |
| H4: PE→AB | 0.157 | 2.377 | Supported |
| H5: PE→SN | 0.132 | 1.235 | Not supported |
| H6: PB→PBC | 0.467 | 6.472 | Supported |

The data analysis results of the extended planned behavior theory model are shown in Figure 2.

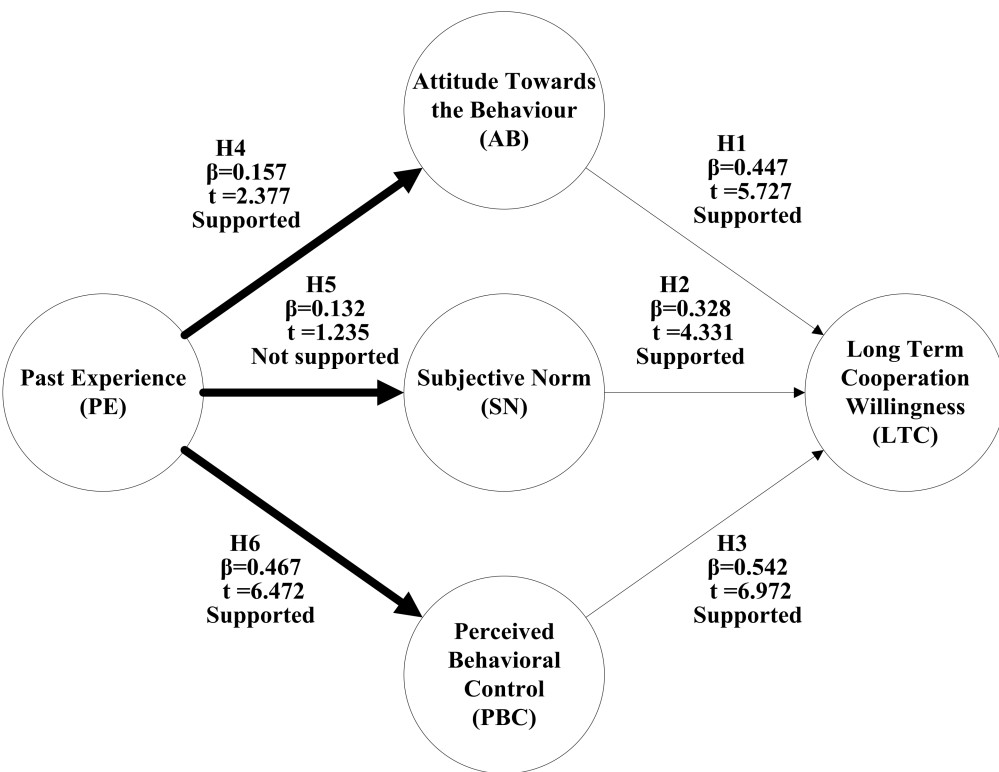

**Figure 2.** Structural equation modeling data analysis results.

## 6. Discussion and Implications

The purpose of this research is to understand general contractors' long-term cooperative intentions with subcontractors and the factors influencing those intentions. The TPB-based integrative conceptual model was developed in order to clarify how subjective and objective elements affect the general contractor's long-term cooperative behavior. In general, the conceptual model accounts fairly well for the long-term cooperative behavior of general contractors. Research hypotheses were fairly validated.

### 6.1. Impacts of Attitude, Subjective Norms and Perceived Behavior Control on Long-Term Cooperation Intention

The results of hypothesis testing (see Table 6) reveal that attitude ($\beta = 0.447$, t = 5.727), subjective norm ($\beta = 0.328$, t = 4.331) and perceived behavioral control ($\beta = 0.542$, t = 6.972) go hand in hand with the general contractors' long-term cooperative intention positively and significantly, supporting H1, H2 and H3. This result is consistent with the findings of Xie et al. [58], who applied the TPB to optimize a healthy building-rating system, thereby emphasizing the importance of attitudes, subjective norms, and perceived behavioral control on staff health-related behaviors. This finding suggests that increases in the characteristics of attitudes, subjective norms, and perceived behavioral control may increase the likelihood of long-term general contractor cooperation. This finding suggests that increasing characteristics such as attitude, subjective norm, and perceived behavioral control may enhance the likelihood of the general contractor's desire to cooperate over the long run.

As previously predicted, perceived behavior control actively contributes to general contractors' long-term cooperative intention in the context of partnerships between general contractors and their subcontractors. The results of this study support the findings of Zheng et al. that perceived behavioral control facilitates behavioral intentions [78]. The results also show that strong self-perception of contractors' long-term cooperation has a significant influence on their willingness to cooperate with subcontractors. Furthermore, it has been confirmed that perceived behavioral control behavior intention has significant correlation to long-term cooperation with subcontractors. Five items were used to

measure perceptual behavior control, including interface division, division of rights and responsibilities, communication ability and subcontract management reflecting general contractors' main management capability configuration applying to subcontractors. This finding suggests that general contractors should divide subcontract interfaces reasonably and specify responsibilities and rights clearly. In the context of long-term cooperation intention, it requires good communication channels for resolving conflicts between general contractors and subcontractors. Rich management skills and experience are also needed for general contractors.

Consistent with the hypothesis, attitude also significantly influences general contractors' long-term cooperation intention. This is similar to Gloukhovtsev's findings that behavioral attitudes have a significant effect on the intention of ethical consumer behavior [79]. General contractors believing in the value of cooperation and achieving high-quality subcontracting services tend to be more willing to establish long-term cooperation with the subcontractor. Five items were used to measure attitudes, mainly including the general contractor's satisfaction with the subcontractor and the value the subcontractor can provide for the general contractor. Thus, H1 was supported in this context. Hence, the satisfaction, value recognition and trust of general contractors in subcontractors was important.

Subjective norms also positively influence general contractors' long-term cooperation intention. This finding is similar to Phua's study, which concluded that influences from internal firms, peer competitors, and policies can drive contractors to develop a willingness to cooperate [80]. It means that when general contractors determine whether to establish a long-term cooperation relationship with subcontractors, the greater the pressure or expectation from the surrounding environment, the greater the intention to cooperate with subcontractors for the long term. However, analyzing data reveals that subjective norms were less influential than perceptual behavior control and attitudes. Five items were used to measure subjective norms, which mainly include legal norms, policy requirements, owner requirements, and pressures or expectations from company or similar projects. The selection and management of general contractors among subcontractors were restricted by laws and policies, because the construction industry involves many parties and connects with public interest. Subcontractors not only need to obtain qualifications, but also comply with laws and regulations to perform subcontracting behavior. At the same time, general contractors' enterprise systems and the project owner may also put forward requirements regarding the selection of subcontractors. As a result, general contractors would be affected by these factors when they choose long-term cooperation partners.

In summary, perceived behavioral control, attitude and subjective norms all positively influence general contractors' long-term cooperation intention.

### 6.2. Impacts of Past Experience on Attitude, Subjective Norms and Perceived Behavioral Control

Hypothesis testing results show that past experience positively correlates with attitudes ($\beta = 0.157$, t = 2.377) and perceived behavioral control ($\beta = 0.467$, t = 6.472), which verifies hypotheses 4 and 6. Consistent with prior studies [5,23,81], past cooperating experience is an efficient predictor of attitude and perceived behavioral control, which indicates that the more past experience in cooperating there is between a general contractor and subcontractor, the more active the general contractor will be in building a long-term cooperation relationship with subcontractor, and their subjective perception ability to cooperate with subcontractor will also be stronger.

The data analysis shows that past experience has a greater positive role in perceived behavioral control and attitude, which is consistent with previous research findings and fits with the characteristics of the construction industry [19,82]. Two aspects that measure past experience are industry reputation accumulated by subcontractors and the general mutual impression of contractors and subcontractors derived from cooperation in the past. The cooperation between general contractors and subcontractors is a process of continuous run-in and adaptation. It is necessary to constantly balance the needs, goals and capabilities of both sides and constantly revise the communication and coordination mode

between both sides. If general contractors have enough past cooperation experience with subcontractors, the cooperation skills and ability of both sides are bound to be improved, and general contractors will be more active in establishing long-term cooperation with subcontractors too.

However, Hypothesis 5 is not supported by the data analysis results. Past experience has very little bearing on subjective norms ($\beta = 0.132$, $t = 1.235$) which deviates from the original expectation and is inconsistent with similar findings from previous studies [68]. This difference may be due to the fact that in the construction industry, subjective specifications of items are actually objective and include legal requirements, policy specifications, and company systems. Past experience deals more with subjective aspects and has limited influence on these objective factors. Therefore, Hypothesis 5 is not valid.

### 6.3. Theoretical Implications

Two major theoretical contributions of this study are listed below. First, although TPB has been successfully implemented in different contexts, such as bike-sharing, e-learning, and waste management, few studies apply the theory of sustainable cooperative intention relationships. This study utilizes the TPB to comprehensively identify the factors that influence contractors' long-term partnership intention with subcontractors. Second, this study expands the literature on contractors' long-term intention to partner with subcontractors by adding the variable of past experience to take into account the reputation and business capabilities of subcontractors.

### 6.4. Practical Implications

This study presents important practical implications. The results of the study can be used to select the long-term subcontracting partners and encourage the contractor to prepare adequately at the outset and to organize the project as a whole. Hence, the preparatory work is very significant for the final success of the project. At the same time, the terms of the contract can be adjusted flexibly in accordance with the situation of subcontracting, so as to protect their own interests and the success of the project to the greatest extent. Subcontractors, in turn, can better understand the selection preferences of general contractors, and in future cooperation try their best to cater to the selection criteria of general contractors, so as to improve their core competitiveness, provide references and suggestions for themselves to maintain and improve the long-term cooperation intention and enhance the degree of the development of specialization.

### 6.5. Research Deficiencies and Prospects

This study introduces mature planned behavior theory to carry out research on the long-term cooperation intention between general contractors and subcontractors. Although this research has theoretical innovations and practical value, it also contains defects limited by subjective and objective conditions. It is hoped that these shortcomings can be corrected in the follow-up research:(1) Constrained by the respondents to questionnaires, most of the data comes from domestic general contractors, such that the conclusions of this research can explain only the formation mechanism of long-term cooperation intention of Chinese general contracting enterprises. In future research, similar research methods can be used to compare and analyze cases in different countries by collecting data from foreign general contractors and combining those cases with models in the overseas background. (2) Many employees participate in the construction industry in complex environments. In accordance with the specific characteristics of construction, this research extends planned behavior theory and introduces the variable of past experience so as to obtain a good hypothesis verification. However, the model still has room for improvement, which is also an issue that needs to be considered in follow-up research. (3) Long-term cooperation intention of the general contractor may be influenced by factors such as project scale, project complexity, project procurement mode and so on. In follow-up studies, we can view these factors as moderator variables to research and compare research differences in different contexts.

(4) From the point of view of general contractors, this paper researched their long-term cooperation intentions with subcontractors. However, cooperation is the result of mutual negotiation. Despite the fact that the construction sector is a "buyer's market," it is still important to take the subcontractor's intentions into account. Future research should do this better.

## 7. Conclusions

How to enhance the cooperation intention between general contractor and subcontractor and how to improve the relationship between both is an important research proposition in the field of construction. This paper seeks to explain the formation mechanism of their long-term cooperation intention from the perspective of a general contractor and puts forward corresponding suggestions based on research results in the hope of making more contributions to the theory and practice of subcontracting management, promoting long-term partnerships and improving subcontracting management.

Through analyzing data of the extended planned behavior theory model, this research obtains the formation mechanism of the long-term cooperative intention of general contractors, thus further providing theoretical basis for general contractors when they choose long-term partners and suggestions for subcontractors to improve their business. Conclusions from verifying the assumptions of the model are listed below: (1) In the connection between the general contractor and subcontractor, perceived behavioral control has the greatest impact on the general contractor's long-term cooperation intention. The stronger the general contractor's self-awareness of long-term cooperation with subcontractors, including interface division, division of rights and responsibilities, communication ability and subcontract management, the stronger the willingness of the general contractor to establish a long-term cooperation with the subcontractor. (2) Attitude contributes to the general contractor's intention to cooperate in the long term. The more positive the general contractor's evaluation of the partnering subcontractor, the stronger the general contractor's willingness to establish a long-term partnership with the subcontractor. The evaluation includes the general contractor's satisfaction with the subcontractor and the subcontractor's ability to provide a more beneficial value to the general contractor. (3) Subjective norms positively impact the general contractor's long-term cooperation intention. That is, when the general contractor determines whether to build a long-term cooperation with a subcontractor, the greater the pressure or expectation from the surrounding environment—which mainly include legal norms, policy requirements, owner requirements, and pressures or expectations from company or similar projects—the more inclined the general contractor will be to cooperate with the subcontractor for a long time. (4) Past experience, specifically the industry reputation accumulated by the subcontractor and the general impression of the general contractor and subcontractor derived from past cooperation, positively influences attitude and perceived behavioral control, which in turn affects the general contractor's intention to work with the subcontractor in the long term.

**Author Contributions:** Conceptualization, X.L. and D.L.; methodology, X.L. and D.L.; formal analysis, X.L. and D.L.; investigation, D.L. and M.X.; data curation, D.L. and M.X.; writing—original draft preparation, D.L. and M.X.; writing—review and editing, D.L. and M.X.; supervision, X.L. All authors have read and agreed to the published version of the manuscript.

**Funding:** Philosophy and Social Science Research in Colleges and Universities in Jiangsu Province (No. 2020SJA1394), Fundamental Research Funds for the Central Universities (No. 331711105).

**Institutional Review Board Statement:** Ethical review and approval were waived for this study, due to this study not involving biological human experiments and patient data, which was not within the scope of review by the Institutional Review Board of Suzhou University of Science and Technology.

**Informed Consent Statement:** Informed consent was obtained from all subjects involved in the study.

**Data Availability Statement:** The data presented in this study are available on request from the corresponding author.

**Acknowledgments:** The authors would like to appreciate the reviewers for all their helpful comments, and to thank the foundation of Philosophy and Social Science Research in Colleges and Universities in Jiangsu Province (No. 2020SJA1394), Fundamental Research Funds for the Central Universities (No. 331711105), and the Jiangsu Province Joint Education Program High-Standard Example Project, for their support.

**Conflicts of Interest:** The authors declare no conflict of interest.

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
