# Peer review of "An Extended Theory of Planned Behavior to Explain General Contractors’ Long-Term Cooperation Intentions in Construction Projects: Empirical Evidence from China"

_sustainability, doi:10.3390/su15097072_

Round 1

Reviewer 1 Report

Abstract: 
1. The author only describes the current reality of construction and development, has this been studied in the existing literature? If so, what are the limitations and shortcomings? If not, what is the contribution of this article?

2. The research methodology is not specified.

3. Please briefly summarize the theoretical and practical significance of this research.

Introduction:

4. What do EPC and BOT mean? The article has the problem of not explaining the meaning of the abbreviations when they first appear in many places.

5. The scientific questions that are not clearly presented are not addressed.

Literature Review:

6. The sentence "Long-term cooperation intention refers to …. contractor and subcontractor" has no logical relation to the context. It is suggested to explain the sentence after "the general contractor will have long-term cooperation intention when he is satisfied with some subcontractors".

7. After explaining the three relationships between general contractor and subcontractor, it is suggested that the author explain again the reasons why he does not choose the other two and focuses on long-term cooperative relationship, so as to highlight that the research perspective of this paper conforms to the policy and practical background.

8. 3.1 section are the original variables to explain the theory of planned behavior. AB, PBC, SN and long-term cooperation intention are not at the same level, so it is not appropriate to put them in this part.

Methodology:

9. The subjects of the questionnaire were limited to contractors in Jiangsu Province, and the number of questionnaires was too small, with only 165 valid questionnaires

Are the conclusions not applicable to the general contractors in China?

10. This paper uses structural equation model for empirical research, but the author does not introduce this method. Please supplement the advantages and disadvantages of this method, its application scope and the reasons for choosing it

Data analysis:

11. Authoritative literature and standards should be added to support the results of each test item (Cronbach 'α, AVE, CE, etc.) in order to meet the requirements.

Discussion and Implications:

12. When analyzing and discussing the test results, the author only said "it is consistent with the previous research results", but did not combine specific literature and specific analysis, which is not convincing enough.

13. Hypothesis 5 is not valid and inconsistent with previous similar research results, but the author does not give a reasonable explanation for this.

Conclusions:

14. Since this paper starts from the perspective of general contractor, it should provide corresponding management and decision-making suggestions to general contractor enterprises one by one according to the conclusion.

Author Response

Response to the Editor and Reviewers’ Comments

Manuscript ID: sustainability-2282809

Title: " An extended theory of planned behavior to explain general contractors' long-term cooperation intentions in construction projects: empirical evidence from China "

Author(s): XUN LIU*, DEXIN LIU, MENGYU XU

Revision due before: 3-Apr-2023

We appreciate the time and effort of the Reviewers and the Editor in reviewing our manuscript. The reviews are very helpful for us to improve the manuscript. Because of the comments from both the Editor and the Reviewers, we have made significant changes and have rewritten parts of the manuscript. Point to point responses to all comments are as follows. The revised contents were shown in red color in the revised manuscript.

Reviewer: 1

Comments:

  1. The author only describes the current reality of construction and development, has this been studied in the existing literature? If so, what are the limitations and shortcomings? If not, what is the contribution of this article?

Response:

We appreciate your comments, we have rewritten the abstract to make it more clear and concise.

“With the continuous development of the construction industry, the current construction production mode is gradually transformed into the contractor, professional subcontractor and labor subcontractor's “main-sub contractor” cooperative production mode in the market segmentation. Long-term cooperation between contractors and subcontractors is beneficial to both parties, yet limited research has explored the factors that determine contractors' willingness to cooperate with subcontractors on a long-term basis. This paper explores the factors influencing the long-term cooperation intention of the contractor and the subcontractor, and the forming mechanism of the long-term cooperation intention of the contractor. To achieve this goal, this study combines the characteristics of the construction industry to increase the variable of past experience, constructing an extended planned behavior theory model. Through questionnaire surveys, contractors with experience in subcontract management were surveyed , and the hypotheses were tested using structural equation modeling techniques. The results showed that contractors' long-term cooperation intention was mainly influenced by attitudes, subjective norms, and perceived behavioral control. While past experience, a new variable, had significant effects on attitudes and perceived behavioral control, influencing contractors' long-term cooperation intention by affecting attitudes and perceived behavioral control, past experience had no direct effect on subjective norms. This study well explain the formation mechanism of the general contractor and subcontractor’s long-term cooperative relationship, and provide a theoretical basis for the general contractor to select partners and suggestions for subcontractors to improve their work.”

Comments:

  1. The research methodology is not specified.

Response:

Thank you for your valuable comments. We have made changes to the abstract section as suggested to make it clearer.

Comments:

  1. Please briefly summarize the theoretical and practical significance of this research.

Response:

Thanks to your comments. We have briefly summarized the theoretical and practical implications of this study at the end of the abstract.

Comments:

  1. What do EPC and BOT mean? The article has the problem of not explaining the meaning of the abbreviations when they first appear in many places.

Response:

We have corrected abbreviations in the article, such as “ Increasingly, Engineering Procurement Construction (EPC) and Build-Operate-Transfer (BOT) projects are being undertaken on the market ”.

Comments:

  1. The scientific questions that are not clearly presented are not addressed.

Response:

Thanks for your valuable suggestions. The introduction section has been revised to clearly present the research questions and emphasize the importance of the study. The specific modifications are as follows:

“The importance of the relationship between general contractor and subcontractor in engineering projects is becoming more and more prominent, and many scholars have studied it. However, most studies focus on the cooperative behavior of subcontractors [1,2], while it is important to note that the cooperative behavior is largely determined by the cooperative intention. Therefore it is necessary to pay attention to the long-term cooperation intention of general contractors to subcontractors. Since general contractor plays a dominant role in general contractor and subcontractor’s sustainable cooperative relationship, this study intends to stand on the general contractors' position to carry out. As for the initial cooperation of general contractor and subcontractor, past experiences involving reputation and workability will provide more information and basis for the general contractor to select subcontractors. [3, 4]. Using the extended theory of planned behavior (TPB) as a theoretical framework, this study explores what factors affect the contractor's choice of long-term cooperation and how to achieve the cooperation intention with subcontractors [5, 6].”

References:

[1] Zhang S, Fu Y, Kang F. How to foster contractors' cooperative behavior in the Chinese construction industry: Direct and interaction effects of power and contract[J]. International journal of project management, 2018, 36(7): 940-953.

[2] Song H, Hou J, Tang S. From contractual flexibility to contractor's cooperative behavior in construction projects: The multiple mediation effects of ongoing trust and justice perception[J]. Sustainability, 2021, 13(24): 13654.

[3] Luhr, G.J., M.G.C. Bosch-Rekveldt, and M. Radujkovic, Key stakeholders' perspectives on the ideal partnering culture in construction projects. Frontiers of Engineering Management, 2020.

[4] Lee, H.-s., et al., Transaction-Cost-Based Selection of Appropriate General Contractor-Subcontractor Relationship Type. Journal of Construction Engineering and Management, 2009. 135(11): p. 1232-1240.

[5] Ajzen, I. and M. Fishbein, Questions raised by a reasoned action approach: comment on Ogden. Health Psychology, 2004. 23(4): p. 431-434.

[6] Ajzen, I., The theory of planned behaviour is alive and well, and not ready to retire: a commentary on Sniehotta, Presseau, and Araújo-Soares. Health Psychology Review, 2015. 9(2): p. 131-137.

Comments:

  1. The sentence "Long-term cooperation intention refers to …. contractor and subcontractor" has no logical relation to the context. It is suggested to explain the sentence after "the general contractor will have long-term cooperation intention when he is satisfied with some subcontractors".

Response:

Thanks to your comment, the corresponding sentence has now been changed to:

Nevertheless, if the general contractor is satisfied with the construction performance and the level of cooperation, there will be a long-term intention to cooperate. Long-term cooperation intention refers to the behavioral orientation or ideological tendency of both parties to hold a long-term joint undertaking.

Comments:

  1. After explaining the three relationships between general contractor and subcontractor, it is suggested that the author explain again the reasons why he does not choose the other two and focuses on long-term cooperative relationship, so as to highlight that the research perspective of this paper conforms to the policy and practical background.

Response:

We have highlighted the reasons for focusing on long-term partnerships in this section. The specific changes are as follows:

“Long-term partnerships can improve performance and significantly improve construction project delivery [1]. To begin with, in order to finish each major trade of construction work, such as excavating, drainage laying, formwork, concreting, metalwork, etc., many general contractors work with a cluster of subcontractors with long-term cooperative relationship, which may last for more than 10 years by estimating conservatively.”

References:

[1] Eom, S.-J., S.-C. Kim, and W.-S. Jang, Paradigm shift in main contractor-subcontractor partnerships with an e-procurement framework. Ksce Journal of Civil Engineering, 2015. 19(7): p. 1951-1961.

Comments:

  1. 3.1 section are the original variables to explain the theory of planned behavior. AB, PBC, SN and long-term cooperation intention are not at the same level, so it is not appropriate to put them in this part.

Response:

Based on the reviewers' comments, we restructured the hypothesis and theoretical model sections.

Comments:

  1. The subjects of the questionnaire were limited to contractors in Jiangsu Province, and the number of questionnaires was too small, with only 165 valid questionnaires

Are the conclusions not applicable to the general contractors in China?

Response:

Thank you for your suggestions. The sample size of this study may indeed be inadequate, but not unreasonable. For example, Boomsma (1985) argued that 100–200 samples would be adaptive for SEM models [1]. Moreover, the current sample also has good reliability and validity and passed the test. The results of this study apply to Chinese contractors, taking into account the previous statement that was not rigorous: the questionnaire collectors were in Jiangsu, but the respondents were not limited to Jiangsu. We have modified it: “Between June 2021 and December 2021, 215 questionnaires were collected, of which 165 were considered valid. ”

References:

[1] Boomsma, A. (1985), “Nonconvergence, improper solutions, and starting values in LISREL maximum likelihood estimation”, Psychometrika, Vol. 50 No. 2, pp. 229-242.

Comments:

  1. This paper uses structural equation model for empirical research, but the author does not introduce this method. Please supplement the advantages and disadvantages of this method, its application scope and the reasons for choosing it

Response:

We appreciate your suggestion. In the data analysis section we have added an introduction to structural equation modeling and a brief description of its applicability.The revisions are as follows:

“Structural equation modeling (SEM) is used to test the proposed hypotheses. SEM analysis is a measurement technique that enables complex causal links between sample data to be expressed in terms of the corresponding model equations and to be measured and analyzed. SEM is therefore particularly suitable for testing predictive and extended theoretical models. In this study, a two-step testing approach was used to first assess the reliability and validity of the measurements included in the current study and then test the hypothesized relationships.”

Comments:

  1. Authoritative literature and standards should be added to support the results of each test item (Cronbach 'α, AVE, CE, etc.) in order to meet the requirements.

Response:

We appreciate your comments. We have added authoritative literature for the results of each test.

References:

[1] Fornell, C.; Larcker, D.F. Evaluating Structural Equation Models with Unobservable Variables and Measurement Error. J. Mark. Res. 1981, 18, 39–50. [CrossRef]

[2] Fornell, C. A Second Generation of Multivariate Analysis: Methods; Praeger: Westport, CT, USA, 1982.

Comments:

  1. When analyzing and discussing the test results, the author only said "it is consistent with the previous research results", but did not combine specific literature and specific analysis, which is not convincing enough.

Response:

We thank the reviewers for their suggestions. The experimental results have now been analyzed and discussed in relation to the specific literature. The modifications are as follows:

“This result is consistent with the findings of Xie et al [58], who applied the theory of planned behavior (TPB) to optimize a healthy building rating system, thereby emphasizing the importance of attitudes, subjective norms, and perceived behavioral control on staff health-related behaviors. This finding suggests that increases in the characteristics of attitudes, subjective norms, and perceived behavioral control may increase the likelihood of long-term general contractor cooperation. This finding suggests that increasing characteristics such as attitude, subjective norm, and perceived behavioral control may enhance the likelihood of the general contractor's desire to cooperate over the long run.”

References:

[58] Xie, X.H., R.B. Wang, and Z.H. Gou, Incorporating motivation and execution into healthy building rating systems based on the theory of planned behaviour (TPB). Building and Environment, 2022. 222.

Comments:

  1. Hypothesis 5 is not valid and inconsistent with previous similar research results, but the author does not give a reasonable explanation for this.

Response:

We thank the reviewer’s suggestions. We have added an explanation for the non-validity of hypothesis 5 as requested by the suggestion.

“However, hypothesis 5 is not supported by the data analysis results. Past experience has very little toll on subjective norm (β=0.132, t=1.235) which deviates from the original expectation and inconsistent with similar findings from previous studies. This difference may be due to the fact that in the construction industry, subjective specifications of items are objective, including legal requirements, policy specifications, and company systems. Past experience deals more with subjective aspects and has limited influence on these objective factors. Therefore, hypothesis 5 is not valid.”

Comments:

  1. Since this paper starts from the perspective of general contractor, it should provide corresponding management and decision-making suggestions to general contractor enterprises one by one according to the conclusion.

Response:

We are grateful to the reviewers for their valuable suggestions. Considering the structure of the article, we have placed the corresponding management and decision-making recommendations in the "Discussion and Insights" section. The details are as follows:

“6.3. Research deficiencies and prospects

Two major theoretical contributions of this study are listed below. First, although TPB has been successfully implemented in different contexts, such as bike-sharing, e-learning, and waste management, there are few studies that apply the theory of sustainable cooperative intention relationships. This study utilizes the TPB to comprehensively identify the factors that influence contractors' long-term partnership intention with subcontractors. Second, this study expands the literature on contractors' long-term intention to partner with subcontractors by adding the variable of past experience to take into account the reputation and business capabilities of subcontractors.

6.4. Practical Implications

This study presents important practical implications. The results of the study can be used to select the long-term subcontracting partners, promote the contractor to prepare adequately at the outset and to organize the project as a whole. Hence, it is very significant that the preparatory work to the final success of the project. At the same time, the terms of the contract can be adjusted flexibly in accordance with the situation of subcontracting, so as to protect their own interests and the success of the project to the greatest extent. From the point of subcontractors, subcontractors can better understand the selection preferences of general contractors, and in the future cooperation, try their best to cater to the selection criteria of general contractors, so as to improve their core competitiveness, provide references and suggestions for themselves to maintain and improve the long-term cooperation intention and enhance the degree of the development of specialization.”

Reviewer 2 Report

Interesting topic of study, however, there are some concerns with the study and the manuscript.

1. The study is statistical-based research that used Structural Equation Modelling as the methodology. Based on the model, three moderating/mediating variables influence the relation (main) between experience with long term-cooperation intention. This central concept is not highlighted in the manuscript, the Authors tend to justify the relationship between each variable and the hypotheses that were formed from the developed model. The concept of moderating/mediating variables should be highlighted and elaborated from the problem statement until data analysis and discussion.

2. The problem statement of this research is poorly developed. This study highlights a specific area of study; thus, the problem statement should also be developed from relevant empirical issues. Authors need to highlight significant issues that may be resulted if the concept of the study is not applied.

3. The scope of the study is also not clearly justified, whereby there is no classification/category of a construction project (the scale of the project) that was being studied. The determination of scope will certainly control the appropriateness, significance and validity of the study, particularly in generalising the concept and the result. These details also need to be included in the methodology. Authors have to reveal the experiences of the respondents involved, and in which type and scale of the project the respondent was involved. The construction project of infrastructure is certainly different from building/real estate. During the survey, Authors also have to ensure that all respondents are aware of which type of project they were asked their perception of. Otherwise, it is no point to present the result of the reliability and validity test since it will be just a matter of numbers. 

4. Table 1: the source (references) needs to be specialised (divided based on each indicator). Authors present the source in the group; each variable should have its references. Also, the Authors need to present the result from the pilot study if there are variances between the result of the literature study with the pilot study. 

5. Line 295: Likert scale should be described as an "agreement scale".

6. Subchapter 5.1 and Table 2: add the type and scale of the construction projects experienced by the respondents.

7. The results were poorly discussed and synthesised.

Author Response

Response to the Editor and Reviewers’ Comments

Manuscript ID: sustainability-2282809

Title: " An extended theory of planned behavior to explain general contractors' long-term cooperation intentions in construction projects: empirical evidence from China "

Author(s): XUN LIU*, DEXIN LIU, MENGYU XU

Revision due before: 3-Apr-2023 

We appreciate the time and effort of the Reviewers and the Editor in reviewing our manuscript. The reviews are very helpful for us to improve the manuscript. Because of the comments from both the Editor and the Reviewers, we have made significant changes and have rewritten parts of the manuscript. Point to point responses to all comments are as follows. The revised contents were shown in red color in the revised manuscript.

Reviewer: 2

Comments:

The study is statistical-based research that used Structural Equation Modelling as the methodology. Based on the model, three moderating/mediating variables influence the relation (main) between experience with long term-cooperation intention. This central concept is not highlighted in the manuscript, the Authors tend to justify the relationship between each variable and the hypotheses that were formed from the developed model. The concept of moderating/mediating variables should be highlighted and elaborated from the problem statement until data analysis and discussion.

Response:

We appreciate your suggestions. The purpose of this study is to understand the long-term cooperation intentions of general contractors and subcontractors based on the theory of planned behavior. An extended Theory of Planned Behavior model is also developed by adding the variable of past experience. The focus of this study is to elucidate how the variables achieve their effects on long-term cooperation intentions. We tend to see past experience and classical behavioral theory (including attitudes, subjective norms, and perceived behavioral control) more as a whole, acting together on long-term cooperative intentions. Meanwhile, the empirical studies on the extended theory of planned behavior are similar, focusing more on the effects and less on the moderating/mediating variables [1-3]. Thus there may be less narrative related to moderating/mediating variables.

References:

[1] Aziz F, Md Rami A A, Zaremohzzabieh Z, et al. Effects of emotions and ethics on pro-environmental behavior of university employees: A model based on the theory of planned behavior[J]. Sustainability, 2021, 13(13): 7062.

[2] Dang Q. Research on the impact of media credibility on risk perception of COVID-19 and the sustainable travel intention of Chinese residents based on an extended TPB model in the post-pandemic context[J]. Sustainability, 2022, 14(14): 8729.

[3] Li J, Shen J, Jia B. Exploring intention to use shared electric bicycles by the extended theory of planned behavior[J]. Sustainability, 2021, 13(8): 4137.

Comments:

The problem statement of this research is poorly developed. This study highlights a specific area of study; thus, the problem statement should also be developed from relevant empirical issues. Authors need to highlight significant issues that may be resulted if the concept of the study is not applied.

Response:

Thank you for your valuable suggestions. The introduction section has been revised to clearly present the research questions and emphasize the importance of the study. The specific modifications are as follows:

“The importance of the relationship between general contractor and subcontractor in engineering projects is becoming more and more prominent, and many scholars have studied it. However, most studies focus on the cooperative behavior of subcontractors [1,2], while it is important to note that the cooperative behavior is largely determined by the cooperative intention. Therefore it is necessary to pay attention to the long-term cooperation intention of general contractors to subcontractors. Since general contractor plays a dominant role in general contractor and subcontractor’s sustainable cooperative relationship, this study intends to stand on the general contractors' position to carry out. As for the initial cooperation of general contractor and subcontractor, past experiences involving reputation and workability will provide more information and basis for the general contractor to select subcontractors. [3, 4]. Using the extended theory of planned behavior (TPB) as a theoretical framework, this study explores what factors affect the contractor's choice of long-term cooperation and how to achieve the cooperation intention with subcontractors [5, 6].”

References:

[1] Zhang S, Fu Y, Kang F. How to foster contractors' cooperative behavior in the Chinese construction industry: Direct and interaction effects of power and contract[J]. International journal of project management, 2018, 36(7): 940-953.

[2] Song H, Hou J, Tang S. From contractual flexibility to contractor's cooperative behavior in construction projects: The multiple mediation effects of ongoing trust and justice perception[J]. Sustainability, 2021, 13(24): 13654.

[3] Luhr, G.J., M.G.C. Bosch-Rekveldt, and M. Radujkovic, Key stakeholders' perspectives on the ideal partnering culture in construction projects. Frontiers of Engineering Management, 2020.

[4] Lee, H.-s., et al., Transaction-Cost-Based Selection of Appropriate General Contractor-Subcontractor Relationship Type. Journal of Construction Engineering and Management, 2009. 135(11): p. 1232-1240.

[5] Ajzen, I. and M. Fishbein, Questions raised by a reasoned action approach: comment on Ogden. Health Psychology, 2004. 23(4): p. 431-434.

[6] Ajzen, I., The theory of planned behaviour is alive and well, and not ready to retire: a commentary on Sniehotta, Presseau, and Araújo-Soares. Health Psychology Review, 2015. 9(2): p. 131-137.

Comments:

The scope of the study is also not clearly justified, whereby there is no classification/category of a construction project (the scale of the project) that was being studied. The determination of scope will certainly control the appropriateness, significance and validity of the study, particularly in generalising the concept and the result. These details also need to be included in the methodology. Authors have to reveal the experiences of the respondents involved, and in which type and scale of the project the respondent was involved. The construction project of infrastructure is certainly different from building/real estate. During the survey, Authors also have to ensure that all respondents are aware of which type of project they were asked their perception of. Otherwise, it is no point to present the result of the reliability and validity test since it will be just a matter of numbers.

Response:

Thank you for your valuable comments. The purpose of this study is to explore the factors that influence contractors' willingness to cooperate with subcontractors in the long term and the mechanisms that shape them. During the questionnaire collection phase, we distributed questionnaires mainly to those engaged in infrastructure projects such as water conservancy, municipalities, highways, and railroads. However, this study focuses on the relationship between engineering contractors and subcontractors, i.e., how the willingness to cooperate in the long term is formed. Most of the respondents in this study are managers with extensive project experience. Considering that every type of engineering project involves more or less this partnership, it should not be unfeasible from a perspective that encompasses multiple project types.

Comments:

Table 1: the source (references) needs to be specialised (divided based on each indicator). Authors present the source in the group; each variable should have its references. Also, the Authors need to present the result from the pilot study if there are variances between the result of the literature study with the pilot study.

Response:

Thanks for your suggestions. Since attitudes toward behavior (AB), perceived behavioral control (PBC), and subjective paradigms (SN) are all part of the theory of planned behavior, their measurement items appear together from time to time. Therefore, these three metrics require mostly the same literature to be cited. We have refined the discussion of the study results in the Discussion section.

Comments:

Line 295: Likert scale should be described as an "agreement scale".

Response:

We thank the reviewers for their valuable comments. We have revised this section:

“A five-point Likert scale was used to assess the items, and respondents were asked to give a rating of 1 to 5 on each item (1 = strongly disagree, 3 = agree, and 5 = strongly agree).”

Comments:

Subchapter 5.1 and Table 2: add the type and scale of the construction projects experienced by the respondents.

Response:

We are thankful for your suggestions.

The questionnaire for this study asked respondents to complete it for the most recent project they had been involved in, and there were no statistics on the type and size of the project. Moreover, the starting point of this study is to clarify the willingness of contractors to cooperate with subcontractors, and more attention is given to the analysis of the partnership. Every type of project involves more or less a long-term partnership, so it should not be unfeasible from a perspective that encompasses multiple project types. Thanks again for your suggestion.

Comments:

The results were poorly discussed and synthesised.

Response:

Thank you for your comments. We have enriched the discussion and conclusion sections to make the article fuller in terms of discussion and synthesis.

Reviewer 3 Report

The academic paper under review is well-structured, with a clear intent of the study. The method of using questionnaires is also well-defined. However, the paper has some clunky repetitions that constrain the flow of reading, making it difficult to understand the main notion. Moreover, the methodology chapter lacks clarity regarding the methods used for the literature review, such as the keywords and timespan.

One of the major concerns is the lack of specificity regarding the authors' usage of the term "sustainable cooperation between contractors and sub-contractors" (217). The authors do not provide any criteria against which sustainability can be measured, nor do they clarify if the term "sustainable" can be replaced by "efficient."

In addition, the inconsistent spelling in either American or British English is another issue that needs to be addressed. Many sentences lack meaning or need to be rewritten, such as lines 79, 169, 177, 178, and 182.

There are also several other comments that the authors need to address, including the need to specify what BOT and EPC mean before using the acronymous (lines 47-48), clarify what "main-part" means (line 57), and use the term "client" instead of "owner" for clarity (line 118). The sentence on line 194 needs further clarification, and the authors should use gender-neutral pronouns such as "they" when referring to a person in line 226 and 227.

Finally, the authors should rewrite the sentence on line 254 to make it more rigorous, and provide a clearer definition of the term "impressive" in the context of line 288.

In conclusion, while the academic paper has a well-defined intent and methodology, there are several areas that need to be improved to enhance the clarity and flow of the reading. The authors should pay closer attention to the language used, and provide more specific definitions where needed.

Author Response

Response to the Editor and Reviewers’ Comments

Manuscript ID: sustainability-2282809

Title: " An extended theory of planned behavior to explain general contractors' long-term cooperation intentions in construction projects: empirical evidence from China "

Author(s): XUN LIU*, DEXIN LIU, MENGYU XU

Revision due before: 3-Apr-2023

We appreciate the time and effort of the Reviewers and the Editor in reviewing our manuscript. The reviews are very helpful for us to improve the manuscript. Because of the comments from both the Editor and the Reviewers, we have made significant changes and have rewritten parts of the manuscript. Point to point responses to all comments are as follows. The revised contents were shown in red color in the revised manuscript.

Reviewer: 3

Comments:

The academic paper under review is well-structured, with a clear intent of the study. The method of using questionnaires is also well-defined. However, the paper has some clunky repetitions that constrain the flow of reading, making it difficult to understand the main notion. Moreover, the methodology chapter lacks clarity regarding the methods used for the literature review, such as the keywords and timespan.

Response:

Thank you for your suggestions. We have made adjustments to the literature review section to make it more keyword focused.

Comments:

One of the major concerns is the lack of specificity regarding the authors' usage of the term "sustainable cooperation between contractors and sub-contractors" (217). The authors do not provide any criteria against which sustainability can be measured, nor do they clarify if the term "sustainable" can be replaced by "efficient."

Response:

We are thankful for your comments. This study aims to highlight the mechanisms of formation of long-term cooperation intentions of general contractors. We consider long-term cooperation as a long-lasting and stable relationship, implying continuous cooperation between contractors and subcontractors [1]. As suggested, we have highlighted this concept in the literature review section.

References:

[1] Relationships between main contractors and subcontractors and their impacts on main contractor competitiveness: An empirical study in Hong Kong.

Comments:

In addition, the inconsistent spelling in either American or British English is another issue that needs to be addressed. Many sentences lack meaning or need to be rewritten, such as lines 79, 169, 177, 178, and 182.

Response:

Thank you for your comments, we have rewritten the sentences to make them clearer and more concise. The specific modifications are as follows.

Lines 79:

Using the extended theory of planned behavior (TPB) as a theoretical framework, this study explores what factors affect the contractor's choice of long-term cooperation and how to achieve the cooperation intention with subcontractors.

Lines 169:

Some scholars have conducted empirical validation, such as green behavior, online shopping behavior, product choice, and eating behavior, which have been well verified. It shows that empirical studies have good predictive power for people's intention and behavior.

Lines 177 and 178:

The introduction of two variables, emotion and moral perception, gives the extended theory of planned behavior greater predictive power.

Lines 182:

The theoretical model is extended by adding different measured variables that fit the model context. In this way, the predictive power of the model is improved, as well as the degree of explanation for intentions and behaviors.

Comments:

There are also several other comments that the authors need to address, including the need to specify what BOT and EPC mean before using the acronymous (lines 47-48), clarify what "main-part" means (line 57), and use the term "client" instead of "owner" for clarity (line 118). The sentence on line 194 needs further clarification, and the authors should use gender-neutral pronouns such as "they" when referring to a person in line 226 and 227.

Response:

We have incorporated the relevant contents in our manuscript. The specific modifications are as follows.

Lines 47-48:

Engineering Procurement Construction (EPC) and Build-Operate-Transfer (BOT) projects are being undertaken on the market.

Lines 57:

Considering that the content of the sentence in this line is duplicated in the previous sentence, it is modified as follows: “For projects with complex construction techniques and high requirements for professional level, contractors may not have the corresponding construction capabilities. Thus they need to invest a lot of manpower and material resources, and the ‘input-output’ ratio is not high.”

Lines 118:

The latter is responsible for (1) managing subcontractors and ensuring that their construction operations satisfy the specifications of the client, as well as (2) coordinating with a group of subcontractors.

Lines 194:

Attitude toward behavior involves the positive or negative evaluation of an individual for performing a particular behavior.

Lines 226-227:

PBC is an unconscious factor reflecting perceptions of how comfortable or difficult it is to put a particular behavior into practice. In general, the simpler the behavior appears to be compared to individual ability, the more likely they are to engage in it. In contrast, the more difficult the comparison, the less likely they are to engage in such a behavior.

Comments:

Finally, the authors should rewrite the sentence on line 254 to make it more rigorous, and provide a clearer definition of the term "impressive" in the context of line 288.

Response:

Following your comment, we have rewritten the sentence as follows:

Lines 254:

Ajzen showed that two points should be noted in the extension of the theory of planned behavior: (1) the newly added variables should be clearly defined and distinguished from the original variables, and (2) the newly added variables should be matched with the original variables.

Considering that when the questionnaire was distributed, respondents were asked to answer the questions based on the most recent project they had participated in, line 288 is now changed to the following: “Respondents were asked to answer the questionnaire with the background of the subcontractor in the most recent project they were involved in.”

Comments:

In conclusion, while the academic paper has a well-defined intent and methodology, there are several areas that need to be improved to enhance the clarity and flow of the reading. The authors should pay closer attention to the language used, and provide more specific definitions where needed.

Response:

Thank you again for your comments. In response to your question, we have revised and refined the language of the entire text to make it clearer and more fluent. Also the description of the definition has been revised to make it g more specific.

Round 2

Reviewer 1 Report

The authors have carefully revised the manuscript and the present version is acceptable.

Author Response

Response to the Editor and Reviewers’ Comments

Manuscript ID: sustainability-2282809

Title: " An extended theory of planned behavior to explain general contractors' long-term cooperation intentions in construction projects: empirical evidence from China "

Author(s): XUN LIU*, DEXIN LIU, MENGYU XU

Revision due before: 16-Apr-2023

We appreciate the time and effort of the Reviewers and the Editor in reviewing our manuscript. The reviews are very helpful for us to improve the manuscript. Because of the comments from both the Editor and the Reviewers, we have made significant changes and have rewritten parts of the manuscript. Point to point responses to all comments are as follows. The revised contents were shown in red color in the revised manuscript.

Reviewer: 1

Comments:

The authors have carefully revised the manuscript and the present version is acceptable.

Response:

Thank you very much for your recognition of our work, and we wish you a happy life.

Reviewer 3 Report

The paper improved. Well-done.

Author Response

Response to the Editor and Reviewers’ Comments

Manuscript ID: sustainability-2282809

Title: " An extended theory of planned behavior to explain general contractors' long-term cooperation intentions in construction projects: empirical evidence from China "

Author(s): XUN LIU*, DEXIN LIU, MENGYU XU

Revision due before: 16-Apr-2023

We appreciate the time and effort of the Reviewers and the Editor in reviewing our manuscript. The reviews are very helpful for us to improve the manuscript. Because of the comments from both the Editor and the Reviewers, we have made significant changes and have rewritten parts of the manuscript. Point to point responses to all comments are as follows. The revised contents were shown in red color in the revised manuscript.

Reviewer: 1

Comments:

The paper improved. Well-done.

Response:

Thank you very much for your recognition of our work, and we wish you a happy life.
